

# The crucial representation of deep convection for the cyclogenesis of medicane Ianos

Florian Pantillon[1], Silvio Davolio[2,3], Elenio Avolio[4], Carlos Calvo-Sancho[5], Diego S Carrió[6], Stavros Dafis[7], Emmanouil Flaounas[8], Emanuele Silvio Gentile[9], Juan Jesus Gonzalez-Aleman[10], Suzanne Gray[11], Mario Marcello Miglietta[12,13], Platon Patlakas[14], Ioannis Pytharoulis[15], Didier Ricard[16], Antonio Ricchi[17], and Claudio Sanchez[18]

[1]Laboratoire d'Aérologie, Université de Toulouse, CNRS, UPS, IRD, Toulouse, France
[2]Department of Earth Sciences, Università degli Studi di Milano, Milan, Italy
[3]Institute of Atmospheric Sciences and Climate, National Research Council of Italy, CNR-ISAC, Bologna, Italy
[4]Institute of Atmospheric Sciences and Climate, National Research Council of Italy, CNR-ISAC, Lamezia Terme, Italy
[5]Department of Applied Mathematics, Faculty of Computer Engineering, Universidad de Valladolid, Spain
[6]Meteorology Group, Department of Physics, University of the Balearic Islands, Palma, Spain
[7]National Observatory of Athens, Institute for Environmental Research and Sustainable Development, Athens, Greece
[8]Institute for Atmospheric and Climate Science, ETH, Zurich, Switzerland
[9]Program in Atmospheric and Oceanic Sciences, Princeton University, Princeton, New Jersey, United States
[10]Department of Development and Applications, Spanish State Meteorological Agency, AEMET, Madrid, Spain
[11]Department of Meteorology, University of Reading, Reading, United Kingdom
[12]Department of Earth and Geoenvironmental Sciences, University of Bari, Bari, Italy
[13]Institute of Atmospheric Sciences and Climate, National Research Council of Italy, CNR-ISAC, Padua, Italy
[14]Department of Physics, National and Kapodistrian University of Athens, Athens, Greece
[15]Department of Meteorology and Climatology, School of Geology, Aristotle University of Thessaloniki, Thessaloniki, Greece
[16]Centre National de Recherches Météorologiques, Université de Toulouse, Météo-France, CNRS, Toulouse, France
[17]Department of Physical and Chemical Sciences/CETEMPS, University of L'Aquila, L'Aquila, Italy
[18]MetOffice, Exeter, United Kingdom

**Correspondence:** Florian Pantillon (florian.pantillon@aero.obs-mip.fr)

**Abstract.** The paper presents a model intercomparison study to improve the prediction and understanding of Mediterranean cyclone dynamics. It is based on a collective effort with five mesoscale models to look for a robust response among ten numerical frameworks used in the community involved in the networking activity of the EU COST Action "MedCyclones". The obtained multi-model, multi-physics ensemble is applied to the high-impact medicane Ianos of September 2020 with focus

5    on the cyclogenesis phase, which was poorly forecast by numerical weather prediction systems. Models systematically perform better when initialised from operational IFS analysis data compared to the widely used ERA5 reanalysis. Reducing horizontal grid spacing from 10 km with parameterised convection to convection-permitting 2 km further improves the cyclone track and intensity. This highlights the critical role of deep convection during the early development stage. Higher resolution enhances convective activity, which improves the phasing of the cyclone with an upper-level jet and its subsequent intensification and

10    evolution. This upscale impact of convection matches a conceptual model of upscale error growth in the midlatitudes, while it emphasises the crucial interplay between convective and baroclinic processes during medicane cyclogenesis. The ten numerical





frameworks show robust agreement but also reveal model specifics that should be taken into consideration, such as the need for
a parameterization of deep convection even at 2 km horizontal grid spacing in some models. While they require generalisation
to other cases of Mediterranean cyclones, the results provide guidance for the next generation of global convection-permitting
models in weather and climate.

## 1 Introduction

Cyclones are essential elements of the climate system and water cycle in the densely populated Mediterranean basin (Flaounas
et al., 2022), which is known as a hotspot for climate change due to identified increase in water scarcity, droughts and coastal
risks (Ali et al., 2022). However, cyclones are also responsible for climate extremes such as heavy precipitation (Khodayar et al.,
2021), windstorms (Nissen et al., 2010) and dust storms (Flaounas et al., 2015). An accurate representation of Mediterranean
cyclones in numerical models is thus crucial to prepare for their societal impacts, from weather predictions a few days ahead
to climate projections several decades ahead.

Mediterranean cyclone dynamics are largely controlled by the large-scale extratropical circulation. Rossby wave breaking
at the end of the North Atlantic storm track typically initiates cyclogenesis through the elongation of a trough that triggers
baroclinic interaction over the Mediterranean (Scherrmann et al., 2024). Regional specifics such as orographic forcing from
the surrounding mountain ranges (Buzzi et al., 2020) and heat fluxes from the warm sea often contribute to the intensification.
Most cases follow a classical extratropical life cycle involving contributions from both dry dynamic and moist diabatic forcings
(Flaounas et al., 2021). While the immense majority of Mediterranean cyclones and associated hazards present typical synoptic
and mesoscale features of the midlatitudes (e.g., Flaounas et al., 2015; Raveh-Rubin and Wernli, 2016; Davolio et al., 2020;
Lfarh et al., 2023), some rare cases have retained attention of the scientific community and broader public due to their peculiar
appearance bearing resemblance with tropical cyclones in satellite imagery. These cases show tropical traits such as a symmet-
ric structure without surface fronts, a cloud-free centre resembling an eye, and a warm core extending at least partly through
the troposphere. They are widely known as tropical-like cyclones or medicanes (Mediterranean hurricanes; Emanuel, 2005),
although their exact definition is still a matter of intense discussion. One debated topic is the extent to which deep convection
contributes to the formation of a warm core structure with a central eye as in tropical cyclones, rather than the dry intrusion and
warm seclusion characteristic of extratropical cyclones (Fita and Flaounas, 2018; Miglietta and Rotunno, 2019; Dafis et al.,
2020).

Due to their intriguing dynamics and sometimes dramatic consequences, medicanes have been the subject of numerous
modelling studies in recent years investigating the different scales and processes involved. The studies generally agree on an
initiation by baroclinic interaction with an upper-level trough or potential vorticity (PV) streamer, as for other Mediterranean
cyclones, before a strengthening phase through convective activity and eventual transition into a tropical-like cyclone (e.g.,
Miglietta et al., 2017). On large scales, forecasting the Rossby wave dynamics along the North Atlantic wave guide that create
a favourable environment for medicane cyclogenesis has been shown to be challenging a few days ahead (Pantillon et al.,
2013; Di Muzio et al., 2019; Portmann et al., 2020). Locally, the modelled cyclone evolution can depend on subgrid-scale



parameterizations for diabatic processes such as deep convection and cloud microphysics (Miglietta et al., 2015; Pytharoulis et al., 2018) or air-sea exchanges (Tous et al., 2013; Akhtar et al., 2014). In some cases, predicting the intensification is sensitive to the representation of the interaction between PV anomalies from Rossby wave breaking at upper levels and from convection at lower levels (Chaboureau et al., 2012; Cioni et al., 2018; Saraceni et al., 2023).

The analogy with tropical cyclones suggests that the numerical modelling of medicanes may benefit from an accurate rep-
resentation of deep convection. While some studies satisfactorily reproduced a series of medicanes in simulations with param-eterised convection at horizontal grid spacings close to 10 km (Tous et al., 2013; Akhtar et al., 2014; Miglietta et al., 2017), convection-permitting simulations with grid spacing of 2–4 km were successful at capturing, for instance, the September 2006 medicane that was missed by lower-resolution operational forecasts (Moscatello et al., 2008; Chaboureau et al., 2012). Similarly, the challenging prediction of the track of medicane Qendresa in November 2014 clearly improved in convection-
permitting simulations with grid spacing smaller than 5 km (Cioni et al., 2018) and was also found sensitive to the representation of convection in simulations with 7.5 km grid spacing (Pytharoulis et al., 2018). In contrast, for the same case, the large-scale dynamics were found dominant compared to local diabatic processes such as latent heat release and surface heat fluxes in other convection-permitting simulations with 2.5–4 km grid spacing (Carrió et al., 2017; Bouin and Lebeaupin Brossier, 2020). Al-together, the results suggest variability between cases studies but also, to some extent, sensitivity to the modelling framework,
as three different atmospheric models were used in the above mentioned studies related to Qendresa.

Acknowledging the lack of clear insight into the most appropriate numerical configuration, the scope of this work is to conduct a model intercomparison project using a common simulation protocol to allow distinguishing between robust results, which are general among different models of the community, and specific results, which depend on the exact model and configuration. On the one hand, the present work aims at guiding model developments to improve forecasts of Mediterranean
cyclones and simulation of physical processes at play in the basin. On the other hand, it aims at better understanding the dynamics and predictability of Mediterranean cyclogenesis. Five research and operational models are involved, of which several variants are run, building 10 different configurations in total. The choice of models and variants is guided by the involved research groups, who contribute with the numerical framework optimised for their needs, thus providing a multi-model, multi-physics, ad hoc ensemble. Sensitivity tests are applied consistently to all models and variants and are decided as a trade-off
between novelty and wide feasibility in order to obtain robust results.

The chosen case study is Mediterranean cyclone Ianos of September 2020 (Lagouvardos et al., 2022; D'Adderio et al., 2022). Ianos meets several criteria that make it of dynamical and societal relevance, as it developed over high sea surface temperature, involved intense convection, exhibited tropical appearance, and impacted Greece with strong winds, precipitation and storm surges during landfall (Androulidakis et al., 2023; Ferrarin et al., 2023). Furthermore, predicting Ianos was challenging for
operational forecasts. Previous modelling studies showed the influence of the convective parameterization on the cyclone evolution (Saraceni et al., 2023) and of the sea surface temperature on the intensity of Ianos and associated hazards (Varlas et al., 2023). In a companion paper, Sanchez et al. (2023) disentangled the link between sea-surface fluxes and upper-level baroclinic dynamics. They found that the divergent outflow due to preceding precipitation was key to create a favourable large-scale environment for the development of Ianos. Here, the work focuses on the representation of deep convection during the



initial phase of cyclogenesis of Ianos and investigates the added value of km-scale resolution. Other aspects of the diabatic processes at play during the entire life cycle of Ianos, such as air-sea interactions, are addressed in companion papers (Ferrarin et al., 2023; Sanchez et al., 2023).

Section 2 describes the different models and variants involved in the study, as well as the common simulation framework. Section 3 provides a brief overview of the life cycle of Ianos and its predictability in operational forecasts. Section 4 presents

the results of the multi-model, multi-physics approach for control and sensitivity runs. Section 5 analyses the representation of large-scale and convective dynamics in the model runs. Section 6 discusses implications of the work and concludes the paper.

## 2   Methods

Several mesoscale models are run using a common framework, first for control simulations and then for sensitivity tests. Two modelling systems are developed at CNR-ISAC and consist of the hydrostatic model BOLAM (Bologna Limited Area

Model; Buzzi et al., 2003) and the non-hydrostatic, fully compressible, convection-permitting model MOLOCH (Local Model in Hybrid Coordinates; Malguzzi et al., 2006) that in operational practice are nested in a cascade. Meso-NH is the mesoscale non-hydrostatic model of the French research community (Lac et al., 2018) and two variants are used here: one run at Centre National de Recherches Météorologiques (hereafter MESONH-CNRM) and the other one run at Laboratoire d'Aérologie (hereafter MESONH-LAERO). The Met Office Unified Model (MetUM) has got different scientific configurations: the Global

Atmosphere and Land version 7 science configuration (GAL7; Walters et al., 2019) developed for 10–200 km numerical weather prediction and climate global modelling, and the Regional Atmosphere and Land configuration version 2 for mid-latitudes (RAL2M; Bush et al., 2023) developed for km and sub-km scale limited area models over the mid-latitudes. Finally, the WRF (Weather Research and Forecasting) model (Skamarock et al., 2008) is a numerical weather prediction system that solves the fully compressible, non-hydrostatic Euler equations. Five variants of WRF are used in the study: one run at the

Aristotle University of Thessaloniki (hereafter WRF-AUTH), two run at CNR-ISAC (hereafter WRF-ISAC and WRF-ISAC-2), one run at the National Observatory of Athens (hereafter WRF-NOA) and one run at the University of the Balearic Islands (hereafter WRF-UIB). Key characteristics of the models and variants are listed in Table 1, while a detailed description of the physical parameterizations used by each configuration is provided in the Supplementary material.

The models are run on a domain encompassing and as close as possible to the area 28–43°N and 10–25°E (illustrated,

e.g., in Figs. 1 and 2), where the horizontal grid varies slightly depending on the model and projection. At least 50 levels are required in the vertical but the exact number and spacing also depend on the specific model implementation and type of vertical coordinates. The multi-model framework has been applied in Ferrarin et al. (2023) to assess the coastal hazard of Ianos and the associated uncertainty, and the model protocol has been applied in Sanchez et al. (2023) to investigate the role of preceding precipitation in preconditioning the upper-level environment for the development of Ianos.

Control simulations are obtained by initialising the models at 00 UTC on 15 September 2020 and using 6-hourly operational analyses from the Integrated Forecasting System (IFS) of the European Centre for Medium-Range Weather Forecasts (ECMWF) as initial and lateral boundary conditions. The horizontal grid spacing is set to 10 km, which approximately matches





the resolution of IFS analyses and requires parameterization of deep convection. A first sensitivity test is obtained by initialising the models 12 h earlier on 12 UTC 14 September to quantify the influence of the chosen initial time. A second sensitivity test

is obtained by using ECMWF Reanalysis v5 (ERA5), which provides higher frequency (hourly) but lower spatial resolution (about 30 km), as initial and lateral boundary conditions. The ERA5 reanalysis is also produced with the IFS model but with different resolution, data assimilation and model version compared to the operational analysis (see Hersbach et al., 2020, for details). A third sensitivity test is obtained by setting the horizontal grid spacing to 2 km, which allows explicit representation of deep convection (except for BOLAM, see Supplementary material). The higher horizontal resolution is applied on the whole

domain without grid nesting, thus without intermediate model between IFS and the mesoscale run. The vertical resolution and all other parameters are kept identical to the 10 km runs (except for MetUM, see Supplementary material). The configuration of sensitivity tests is summarised in Table 2.

All simulations are run until 00 UTC 19 September, i.e., for 4–4.5 days depending on the initial time, but the focus is on the intensification phase, roughly 15–17 September. On the one hand, the relatively small integration domain and the use

of (re)analyses constrain the large-scale dynamics; on the other hand, the domain is large enough to allow each model to develop its own mesoscale dynamics. Hence, there is no drift from the large scales in the simulations, thus enabling an easier comparison of the mesoscale processes occurring during the cyclone development The output of all model runs is interpolated onto the same regular $0.1° \times 0.1°$ grid and pressure levels to allow a fair comparison.

The simulations are assessed against different data sources. The IFS operational analysis is used as a reference for dynamical

fields. For the track and intensity of Ianos it is complemented by ERA5 reanalyses and also evaluated in the 50 perturbed members of the IFS ensemble prediction system with horizontal resolution of about 18 km. The tracking is performed using a simple algorithm based on the mean-sea-level pressure (MSLP) minimum. While tracking Mediterranean cyclones generally requires a more sophisticated approach (Flaounas et al., 2023), the simple algorithm performs well here thanks to the isolated and well-defined MSLP signature of Ianos over the open sea. Infrared satellite observations from Meteosat Second Generation

(MSG) in the 10.8 $\mu$m channel are used to identify deep convection indicated by cold cloud tops. When available, the Radiative Transfer for TOVS (RTTOV) fast radiative transfer model is applied to simulated fields to compute synthetic MSG brightness temperature observations in the 10.8 $\mu$m infrared channel. Finally, Integrated Multi-satellite Retrievals for GPM (IMERG; Huffman et al., 2023) are used as a reference for precipitation and Advanced Scatterometer (ASCAT) observations for surface wind over sea.

**3 Overview of medicane Ianos**

The cyclogenesis of Ianos takes places in mid September 2020 over the anomalously warm waters of the Gulf of Sidra exceeding 28°C locally (Lagouvardos et al., 2022; Varlas et al., 2023; Sanchez et al., 2023). On 13 September, a convective cluster is present to the south of Sicily associated with a weak cut-off low, while a separated weak surface low is present over Libya (not shown). On 14 September, the convective cluster and the surface low approach each other (Fig. 1a; the presence of dust

reveals the cyclonic circulation on the satellite image). On 15 September, the two structures merge off the Libyan coast but the





convection has weakened (Fig. 1b). On 16 September, the cyclone moves northward and deepens quickly, while the convection redevelops in the north-western quadrant (Fig. 1c). On 17 September, the cyclone attains its mature phase, it exhibits tropical appearance and turns eastward (Fig. 1d). On 18 September, the cyclone makes landfall over Greece, still associated with deep convection (Fig. 1e). Later on, the cyclone bifurcates southward and weakens quickly (Fig. 1f). This short overview suggests

that both pre-existing convection and surface low play a role in the initial cyclogenesis, thus the accurate representation of both may result critical for the successful simulation of the cyclone.

However, as it is usually the case for medicanes, midlatitude dynamics also play a crucial role in the cyclogenesis and development of Ianos. The large-scale environment is characterised by a double jet structure over North Africa with both polar and subtropical jets associated with filaments of PV at upper levels (around 31°N, 14°E and 30°N, 22°E, respectively,

on Fig. 2a). Cyclogenesis is initiated on the poleward side of the polar jet and is surrounded by small PV structures (see first closed MSLP contour near 33°N, 17°E in Fig. 2b). Intensification occurs during the arrival of a PV streamer from the west over the Ionian Sea, which clearly separates a westerly jet on the equatorward side from a southerly jet on the poleward side (Fig. 2c). The PV streamer wraps around the surface cyclone and induces baroclinic interaction on 17 September (Fig. 2d). The system then moves eastward during the mature phase (Fig. 2e) and weakens after landfall (Fig. 2f).

The cyclogenesis of Ianos was poorly forecast a few days ahead, as illustrated in the operational IFS ensemble prediction system initialised prior to intensification (Figure 3). The 50 perturbed ensemble members (grey curves) and the ensemble mean (red curves) are compared to the IFS analysis (black curves) and ERA5 reanalysis (dotted curves) for the track and intensity defined by the minimum MSLP. Forecasts initialised on 14 September 00 UTC show a large spread in position and a relevant southeastward shift of all members compared to the analysed track (Fig. 3a; the red dot marks the mean position

at 00 UTC 17 September), while they largely miss the cyclone intensification (Fig. 3b). Earlier forecasts barely predict the cyclogenesis and are not shown. Forecasts initialised on 15 September 00 UTC perform better but maintain a large spread and a southeastward shift in position, albeit reduced (Fig. 3c). The minimum pressure intensifies below 1000 hPa on average but with increased variability between members, which range from an intense cyclone with 986 hPa minimum MSLP and almost no cyclogenesis with 1008 hPa minimum MSLP (Fig. 3d). Similarly, the IFS analysis and ERA5 reanalysis are affected by large

differences. While they agree well with each other and with satellite observations on the cyclone track, the ERA5 reanalysis indicates MSLP values up to 10 hPa shallower than the IFS analysis during the period of maximum intensity. However, none of forecasts or analyses captures the only MSLP measurement recorded prior to landfall that reaches 984 hPa at the Palliki station in Cephalonia, Greece (blue star on Fig. 3). This suggests that all forecasts and analyses underestimate the actual cyclone intensification. Later forecasts better follow the analysed track and predict a more intense cyclone, although still too shallow

compared to the Palliki record (not shown).

## 4    Model evaluation

Control simulations initialised on 15 September 00 UTC from the IFS analysis are reminiscent of the IFS ensemble members from the same initial time. The simulations also show large spread and southeastward shift in the track of Ianos (Fig. 4a), as





well as large spread in intensity (Fig. 5a), although they generally deepen more than the IFS ensemble and analysis (note the
different scale compared to Fig. 3d). Large spread in track and intensity is found not only between models but also between
the five variants of WRF (dashed curves), which range from the westernmost to the easternmost track and from one of the
deepest to the shallowest cyclone. In contrast, the two variants of MESONH (orange and green curves) well capture both the
analysed track and the single MSLP record at the Palliki station. However, the focus is not on ranking which model performs
better or worse, because a single case study with a specific configuration is not representative of the overall model performance.
Therefore, in the following, the models and variants are not investigated individually but as a whole to emphasise a systematic
response to sensitivity tests. Model specifics are discussed only at the end of the section.

As expected, simulations initialised 12 h earlier perform worse than the control runs. They show similar spread but larger
southeastward shift in the track (Fig. 4b) and stronger underestimation of the intensity (Fig. 5b). This behaviour is again
reminiscent of the operational IFS ensemble prediction system. Additional simulations were initialised at 00 UTC on 14
September but are not shown, because they generally did not develop a cyclone. Initialising the simulations from the ERA5
reanalysis instead of IFS analysis also clearly degrades the results. The tracks are also shifted southeastward (Fig. 4c) with
a bias similar to the earlier initialisation, although with smaller spread. The intensity shows a shallow cyclone and only a
few simulations reach 1000 hPa, far from the actual minimum MSLP, while some do not intensify at all (Fig. 5c). Similar
results were obtained in sensitivity tests mixing ERA5 initial conditions and IFS lateral boundary conditions and confirmed
the prevailing role of the initial conditions (not shown). Increasing horizontal resolution from 10 km to 2 km grid spacing
clearly improves the track and reduces the spread (Fig. 4d). Apart from two outliers that keep a southeastward shift, the
simulations closely follow the analysed northward tracks until 17 September 00 UTC (dots) before deviating southward during
the subsequent eastward motion. This evolution is accompanied by a stronger intensification compared to the control runs
(Fig. 5d). Most simulations produce a deep cyclone during the mature phase on 17 September and with MSLP value even
below the record at landfall on 18 September. The lack of in situ measurements prevents a direct validation of cyclone intensity
during the mature phase.

To complement the MSLP-based evaluation, the simulated intensity is also evaluated in terms of near-surface wind. This
is performed by comparing model wind speed averaged within 1 degree of the simulated cyclone centre (Fig. 6). Several
passages of the ASCAT satellite-borne instrument over Ianos provide measurements near 19 UTC on 15 September, 20 UTC
on 16 September and 19 UTC on 17 September (blue diamonds). The control simulations show progressive increase in wind
speed until landfall, with some spread around measured values (Fig. 6a). Deeper cyclones generally produce stronger winds,
although the link varies between models and variants and likely depends on the surface parameterizations. As for the MSLP,
the wind in the IFS analysis reaches a plateau on 17 September and is underestimated afterwards (black curve), while the
wind in the ERA5 reanalysis is clearly too weak (dotted curve). The wind speed in sensitivity tests is also consistent with the
deepening: early and reanalysis runs produce weaker winds (Fig. 6b,c) whereas high-resolution runs increase their strength
(Fig. 6d). In particular, high-resolution winds are too high compared to ASCAT measurements, which is consistent with the
deeper cyclone compared to the Palliki record (Fig. 5d) and the overestimation of significant wave height in the open sea
resulting from high-resolution runs (Ferrarin et al., 2023). While the saturation of ASCAT retrievals above 20–25 m s$^{-1}$ (Chou





et al., 2013) may produce an underestimation of the actual winds, the apparent overestimation in high-resolution simulations
may be ascribable to an inaccurate representation of the air-sea interactions. In fact, not all the models and variants properly
take into account the increase and saturation in surface roughness due to developing and breaking waves, respectively, or the
cooling of the sea surface, which may have complex and relevant feedbacks on the atmospheric boundary layer and surface
winds (e.g., Gentile et al., 2021).

## 5   Upscale impact of convection

These results highlight the crucial role of both initial conditions and explicit or parameterised representation of deep convection
for the development of Ianos. The reason for the contrasting evolution between sensitivity tests is first investigated analysing
composites of the large-scale dynamics. On 16 September 00 UTC, all composites capture the double jet structure and a weak
surface low located on the poleward side of the polar jet (Fig. 7). However, the surface low better phases with the arriving PV
streamer in the high-resolution (Fig. 7d) than in the control runs (Fig. 7a), while it does not detach from the Libyan coasts
and remains further away from the PV streamer in early and reanalysis runs (Fig. 7b, c; compare with the IFS analysis in
Fig. 2c). Differences in phasing between Ianos and its environment quickly increase and become markedly different during
the next 24 h (Fig. 8). On 17 September 00 UTC, the surface low is located below the polar jet in the control runs (Fig. 8a)
and between the polar and subtropical jets in the early and reanalysis runs (Fig. 8b, c), which all miss the cyclonic wrap-up
of the PV streamer. Only the high-resolution runs correctly predict the baroclinic interaction indicated by the PV streamer
wrapping around the intensifying cyclone (Fig. 8d; compare with Fig. 2d). It is worth noting that the much deeper cyclone
in the high-resolution composite is due to both the deeper cyclone in individual simulations (Fig. 5d) and the lower spread in
cyclone position between the simulations (Fig. 4d).

   While the sensitivity to initial conditions found in early and reanalysis simulations may be related to the representation of
both surface cyclone and large-scale dynamics, the sensitivity to horizontal resolution points mainly to the representation of
deep convection. Sanchez et al. (2023) shows that the outflow related to several bursts of deep convection creates and maintains
a bubble of tropospheric low-PV air at upper levels, which controls the large-scale flow and the development of Ianos. The
first burst is related to a preceding precipitation event that occurs near 00 UTC on 15 September. Its representation cannot
be assessed here, because its timing coincides with the initialisation of all but early simulations. Instead, the focus is shifted
to a second burst of convection that occurs near 00 UTC on 16 September. Infrared satellite observations show extensive
areas of low brightness temperatures at that time, indicating cells of deep convection forming downstream of the surface
cyclone (Fig. 9a). The representation of the second burst of convection is illustrated in high-resolution runs for five models and
variants that provide synthetic observations of brightness temperatures computed from the simulation output (Fig. 9b–f). All
runs develop deep convection but with substantial variability, MESONH-CNRM showing the most extensive convective area,
MESONH-LAERO and MetUM a more moderate convection, and MOLOCH and WRF-AUTH a weaker convective activity.
Control runs develop convective systems of similar spatial extent but lower intensity, while reanalysis runs consistently develop
weaker convection and early runs show a contrasted response depending on the model and variant (not shown).



The representation of deep convection in the same five models and variants around 00 UTC on 16 September is further quantified using the Fractions Skill Score (FSS; Roberts and Lean, 2008; Mittermaier, 2021). The FSS is a spatial verification metric that provides a measure of forecast accuracy at a specific scale and ranges from 0 (no skill) to 1 (perfect forecast skill). It is applied here for two representative scales of 10 and 200 km and for a threshold of 223 K in brightness temperature indicative of deep convection (yellow shading in Fig. 9). While control runs marginally reach the 0.5 benchmark, high-resolution runs stand out and consistently achieve high values of FSS (Fig. 10c, f). This indicates an improved representation of deep convection across scales using horizontal grid spacing of 2 km compared to 10 km. In contrast, early runs degrade its representation (FSS shifted toward lower values in Fig. 10a, d) and reanalysis runs exhibit a shortfall in capturing deep convective activity (FSS values below 0.3 in Fig. 10b, e). This is in line with the previous results and the contrast between sensitivity tests remains consistent with increasing scale.

A more systematic assessment of the representation of convection is undertaken, based on precipitation, which is provided by all models and does not rely on a radiative transfer computation that may introduce additional variability between simulations. The IMERG satellite product exhibits three distinct peaks of precipitation along the track of Ianos (black curves in Fig. 11). The first two peaks near 00 UTC on 15 and 16 September correspond to the two bursts of convection mentioned above, whereas the third peak occurs after 12 UTC on 16 September. The values of $\mathcal{O}(10 \text{ mm } (3 \text{ h})^{-1})$ are averaged within 1 degree from the cyclone centre, and thus denote very intense precipitation. The relatively small radius is chosen to focus on convective precipitation, which is located close to the cyclone centre; rather than stratiform precipitation that is spread over a larger area.

While control runs only partly capture the first peak that occurs near the model initialisation time, they better predict the second peak but largely miss the third and most intense one (Fig. 11a). In contrast, after a similar initial evolution, reanalysis runs clearly underestimate the second peak and generally predict weakening precipitation afterwards (Fig. 11c). The early runs better capture the initial peak thanks to the initialisation before its onset that allows model spin-up (Fig. 11b). However, the representation of the second peak is contrasted in the early runs with one half of the models depicting a weaker peak and the other half a stronger peak compared to the control runs, but all largely missing the third peak. The sensitivity to model resolution is clearer with a similar initial evolution followed by an enhanced second peak and a third peak even overestimated by most high-resolution simulations (Fig. 11d). The strong overestimation of the simulated second peak compared to the observations (Fig. 11d) suggests a systematic bias exists across models or that the IMERG satellite product provides poor estimates of intense convective precipitation. Nevertheless, the subsequent evolution is clearly improved in high-resolution, convection-permitting simulations compared to all convection-parameterised runs (Fig. 8). Thus, intense convection during the second burst around 00 UTC on 16 September is crucial for the phasing between Ianos and the large-scale dynamics, which pilots the cyclone track and intensity via steering and baroclinic interaction. This is consistent with the results of Sanchez et al. (2023), who focus on the interaction between convection and large-scale dynamics during the first burst around 00 UTC on 15 September.

The results are summarised in Fig. 12. Sensitivity tests to the initial data (squares) and horizontal resolution (pluses) compared to the control simulations (circles) show strong relationship in most cases between the location and intensity of Ianos on 17 September 00 UTC: the deeper the simulated cyclone, the closer it is to the analysed position (Fig. 12a). When comparing





the control, reanalysis and high-resolution simulations, a strong relationship is also found on average between the position on 17 September 00 UTC and the intensity of the second convective burst around 16 September 00 UTC: the stronger the precipitation, the smaller the position error 24 h later (Fig. 12b). The relationships are also matched by most of the runs initialised

at 12 UTC on 14 September (triangles) that show larger position error accompanied by a shallower cyclone (Fig. 12a) and preceded by reduced precipitation (Fig. 12b). However, some of the early runs follow an alternative scenario with a cyclone that quickly intensifies close to the North African coast but does not phase with the polar jet and PV streamer at upper levels and slowly moves eastward (not shown). This results in larger position error though deeper cyclone and stronger precipitation compared to the control runs (lines oriented toward the top left in Fig. 12a and toward the top right in Fig. 12b).

Beyond these general relationships from sensitivity tests, the model runs show substantial variability. For instance, the alternative scenario is followed by early WRF simulations only (dashed lines), which suggest a specific model behaviour. The MetUM matches the general relationships well and its precipitation and location values are close to the mean but the cyclone is systematically shallower (red lines). Both MESONH variants perform well but with relatively low sensitivity to the model resolution (orange and green lines), which points the ability of the convection scheme to capture convective bursts. Finally,

MOLOCH also shows relatively low sensitivity to the model resolution (purple lines). This may be related to the absence of a shallow convection scheme and the activation of the deep convection scheme even at 2 km grid spacing. These model-specific results will require generalisation through the analysis of other case studies but have already guided and may further guide the model development.

## 6 Conclusions

The paper presents a model intercomparison study to improve the prediction and understanding of Mediterranean cyclone dynamics. It is based on a collective effort with five mesoscale models to look for a robust response among ten numerical frameworks used in the community involved in the networking activity of the EU COST Action "MedCyclones". The obtained multi-model, multi-physics ensemble is applied to the high-impact medicane Ianos of September 2020 with focus on the cyclogenesis phase, which was poorly forecast by operational numerical weather prediction systems.

The models and variants run with 10 km horizontal grid spacing and convection parameterization show large spread in the track and intensity of Ianos and with magnitude comparable to that of a global ensemble prediction system. However, they react in a consistent fashion to sensitivity tests to initial conditions and horizontal resolution, thus providing a coherent response and a robust indication. Unsurprisingly, an earlier initialisation time degrades the cyclone prediction by increasing the (southeastward) bias in track and underestimation in intensity. Initialising simulations with the ERA5 reanalysis instead

of the operational IFS analysis has a detrimental effect that is similar to a 12 h earlier initialisation on the cyclone track and larger on the intensification, which is hardly captured. In contrast to the other sensitivity tests, increasing horizontal resolution from 10 to 2 km grid spacing clearly improves the cyclone prediction. The error and spread in track are largely reduced and the intensity is closer to and even higher than recorded values, although the maximum cyclone intensity is not well constrained due to the lack of in situ observations and reliable analyses over the open sea.



The contrasted evolution of Ianos simulations is related to its phasing with the large-scale dynamics. While it moves underneath a meandering polar jet in the 10 km runs and between the polar and a subtropical jet when initial conditions are modified, the cyclone remains on the poleward side of the polar jet in the 2 km runs. Only in the latter case the cyclone tracks northward and intensifies in agreement with the actual evolution. The improvement from 10 to 2 km is attributed to the shift from parameterised to explicit representation of deep convection. Indeed, the strength of a convective burst after one day of

simulation is reinforced at 2 km, while it is weakened with modified initial conditions, and appears to constrain the phasing with the upper-level jet one day later and subsequent evolution. Accordingly, Sanchez et al. (2023) shows how several bursts of convection produce and sustain a bubble of low-PV air at upper levels that favourably preconditions the large-scale circulation. Thus, the more realistic representation of deep convection is crucial not only for local effects, such as intense precipitation or latent heat release that may sustain the cyclone during its mature phase, but also for upscale impacts since it critically modifies

the midlatitude circulation in which the cyclone is embedded.

This model behaviour matches the conceptual model of upscale error growth in the midlatitudes by Zhang et al. (2007), in which a first phase of fast error growth associated with convection within a few hours is followed by a second phase of projection of the error on the balanced large-scale dynamics within a few days, before a propagation with large-scale dynamics afterwards. It emphasises the crucial interplay between convective and baroclinic processes during medicane cyclogenesis also

found by other authors (Chaboureau et al., 2012; Miglietta et al., 2015; Cioni et al., 2018; Saraceni et al., 2023). Furthermore, the sensitivity to both initial conditions and model physics suggests a balanced contribution of baroclinic and diabatic processes to the cyclone dynamics as in previous medicanes and other Mediterranean cyclones (Flaounas et al., 2021). However, the results contradict earlier studies that satisfactorily reproduced medicanes with parameterised convection (Tous et al., 2013; Akhtar et al., 2014; Miglietta et al., 2017). Thus, a systematic improvement from convection-permitting resolution is not ex-

pected in general but only for cases where convection plays an important role such as Ianos. This is not restricted to medicanes, as for instance an added value of km-scale resolution was also found for a secondary cyclone (Carrió et al., 2020). Overall, the results may contribute to better understand the specific case of medicanes and contrasts among known cases.

Beyond the robust responses described above, an outcome of this model intercomparison project is to identify the particular behaviour of each model. For instance, MOLOCH shows the need for a convective parameterization even at 2 km (likely due to

the absence of a shallow convection scheme), Meso-NH exhibits weak sensitivity to the resolution possibly due to the formulation of the convection parameterization, and WRF presents an alternative cyclone evolution pattern in some simulations with early initialisation. More generally, the clear improvement with explicit convection provide guidance for the next generation of weather and climate models foreseen at the global scale (e.g., Stevens et al., 2019). Results with the ERA5 reanalysis are noteworthy for weather and climate studies where they are widely used. Indeed, not only does ERA5 clearly underestimate the

cyclone intensity but a mesoscale model downscaling ERA5 from about 30 to 10 km horizontal resolution remains seriously flawed, despite the high temporal resolution availability. This may also concern operational forecasting, as ERA5 is also used to train machine learning–based methods that are currently challenging traditional physics-based global weather forecast models (Bi et al., 2023; Lam et al., 2023). Finally, the developed framework opens the opportunity to test the significance of other processes for Mediterranean cyclone dynamics, such as air-sea interactions and the various fully or partly coupled approaches



used to represent them (e.g., Bouin and Lebeaupin Brossier, 2020; Varlas et al., 2023) or the direct and indirect effects of aerosols that may be accounted for in radiative and cloud microphysics schemes.

*Code and data availability.* ERA5 reanalysis data was downloaded from the Copernicus Climate Change Service https://doi.org/10.24381/ cds.143582cf. IFS analysis data was retrieved from the MARS Catalogue https://apps.ecmwf.int/mars-catalogue (restricted access). The interpolated model data are available in NetCDF format on request. The Meso-NH code is freely available under CeCILL-C license agreement
on http://mesonh.aero.obs-mip.fr/. The BOLAM and MOLOCH codes are available upon request to dinamica@isac.cnr.it. The WRF model is freely available at https://github.com/wrf-model/WRF/releases. The MetUM model is available for use under a closed licence agreement, further information at http://www.metoffice.gov.uk/research/modelling-systems/unified-model.

*Author contributions.* FP and SiD coordinated the model intercomparison; FP, SiD, EA, DSC, StD, MMM, IP, DR and CS provided model data; AR maintained a server to centralise and analyse the data; FP prepared the manuscript and figures, except Fig. 10 that was prepared by
JJGA and CCS; all authors discussed the results and provided feedback on the manuscript.

*Competing interests.* Co-author Silvio Daviolo is a member of the editorial board of WCD.

*Acknowledgements.* This work is a contribution to the COST Action CA19109 "MedCyclones: European Network for Mediterranean Cyclones in weather and climate". We acknowledge the use of imagery from the NASA Worldview application https://worldview.earthdata. nasa.gov/, part of the NASA Earth Observing System Data and Information System (EOSDIS).



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



(a) 0936 UTC 14 Sep 2020    (b) 1156 UTC 15 Sep 2020    (c) 0923 UTC 16 Sep 2020

(d) 1145 UTC 17 Sep 2020    (e) 0910 UTC 18 Sep 2020    (f) 0953 UTC 19 Sep 2020

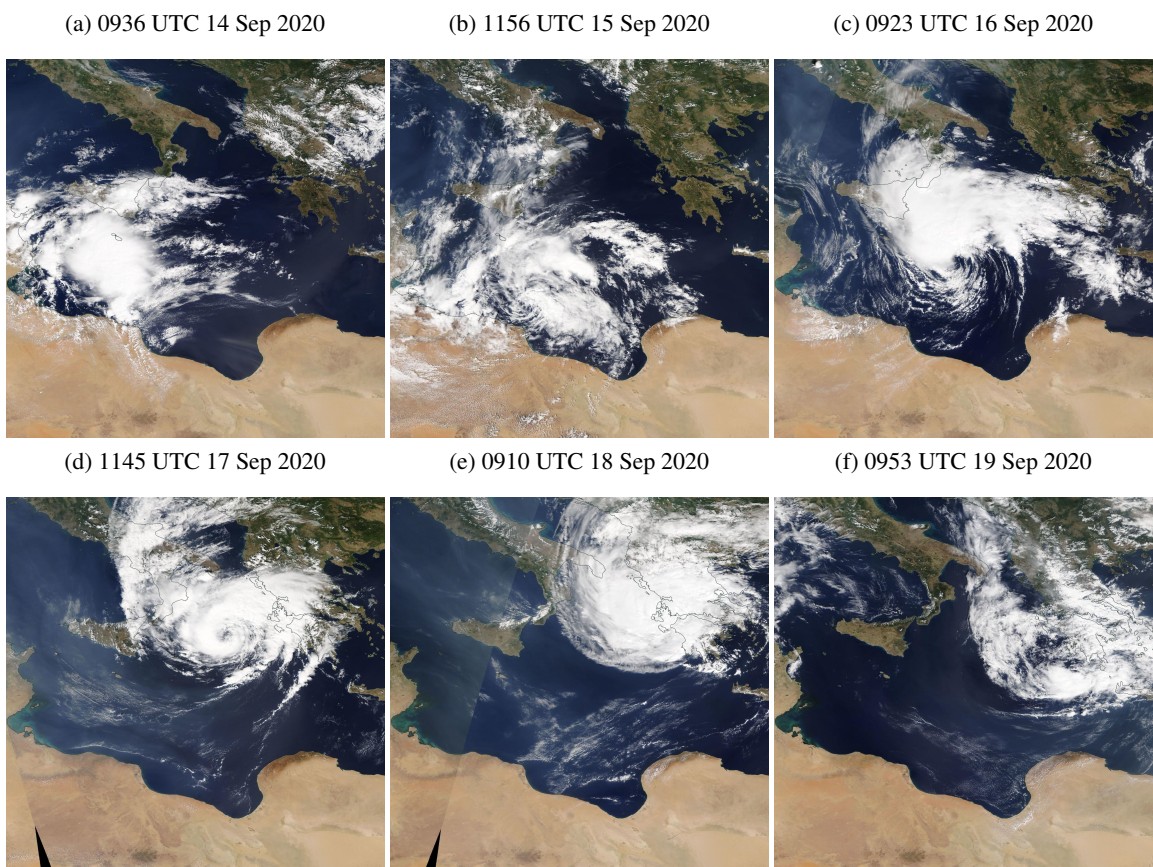

**Figure 1.** Visible satellite observations of Ianos from the Moderate Resolution Imaging Spectroradiometer (MODIS) instrument daily from 14 to 19 September 2020 (a–f). The observation time depends on the passage of the Aqua and Terra satellites.

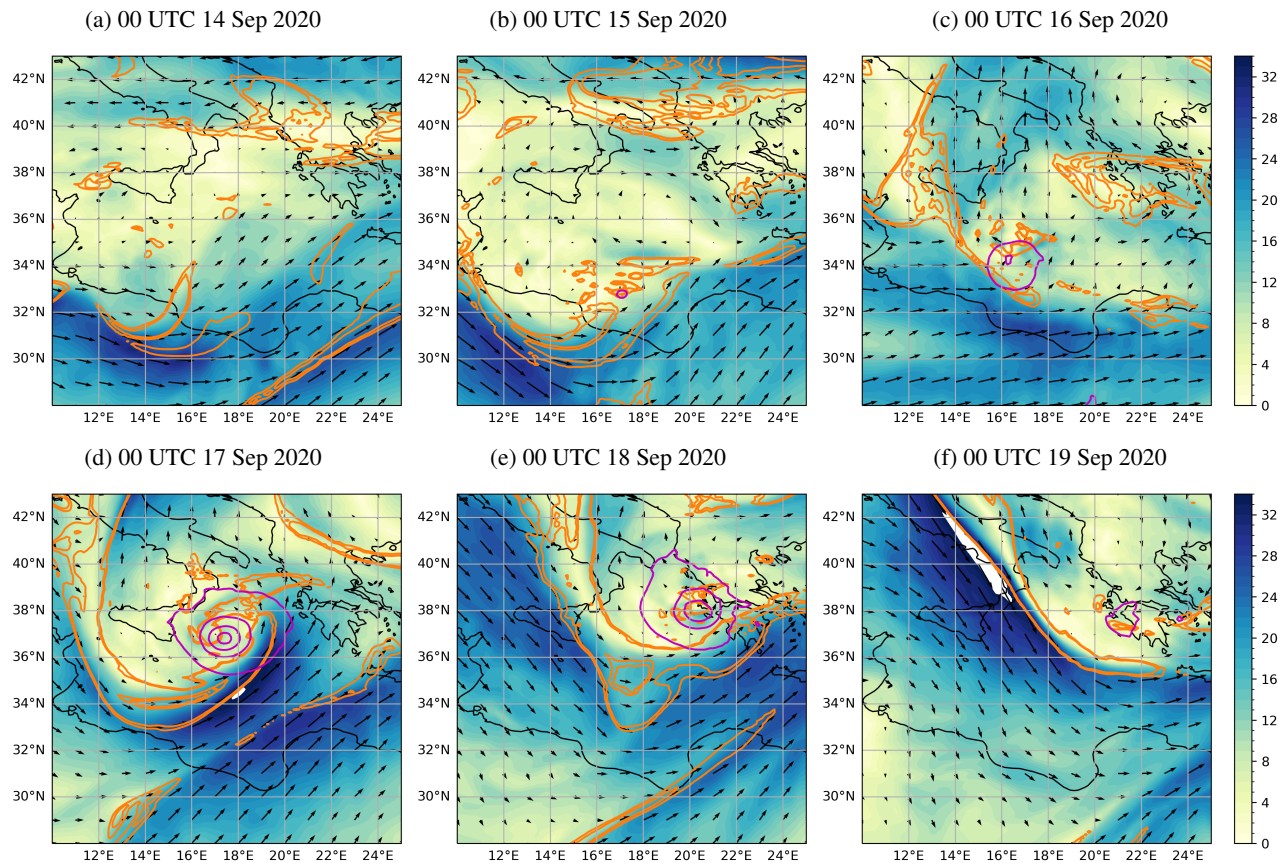

**Figure 2.** Evolution of Ianos in the IFS operational analysis from 14 to 19 September 2020 (a–f): 300 hPa wind (shading in m s$^{-1}$ and vectors), 300 hPa potential vorticity (orange contours at 1.5 and 2 pvu) and MSLP (pink contours every 5 hPa below and including 1010 hPa).

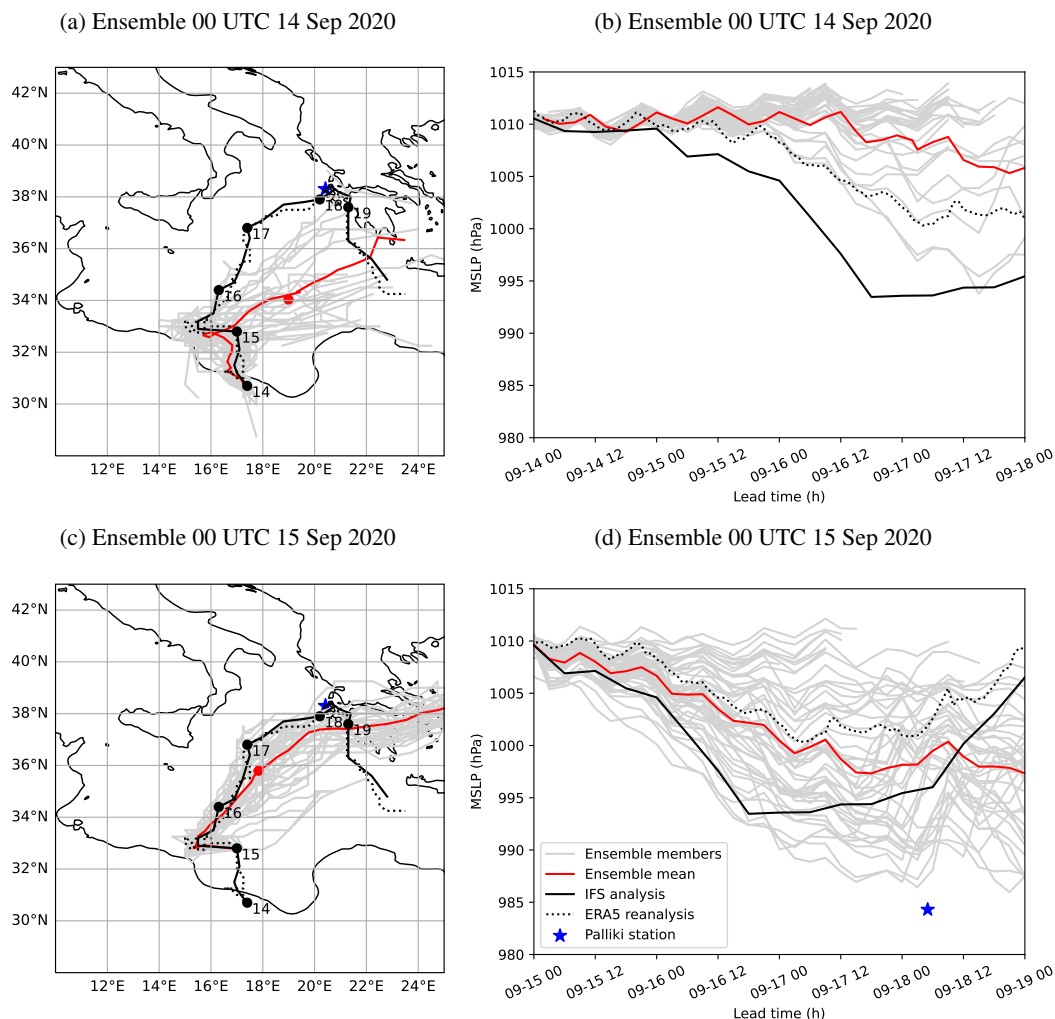

**Figure 3.** Predictability of Ianos in the IFS ensemble prediction system: track (a, c) and intensity (b, d) from 14–18 (a, b) and from 15–19 September 2020 (c, d) in the ensemble members (grey curves) and mean (red curves with dot at 00 UTC 17 September). The IFS analyses (black curves and dots at 00 UTC) and ERA5 reanalyses (dotted curves) are also shown for reference, as well as the station of Palliki (blue star).

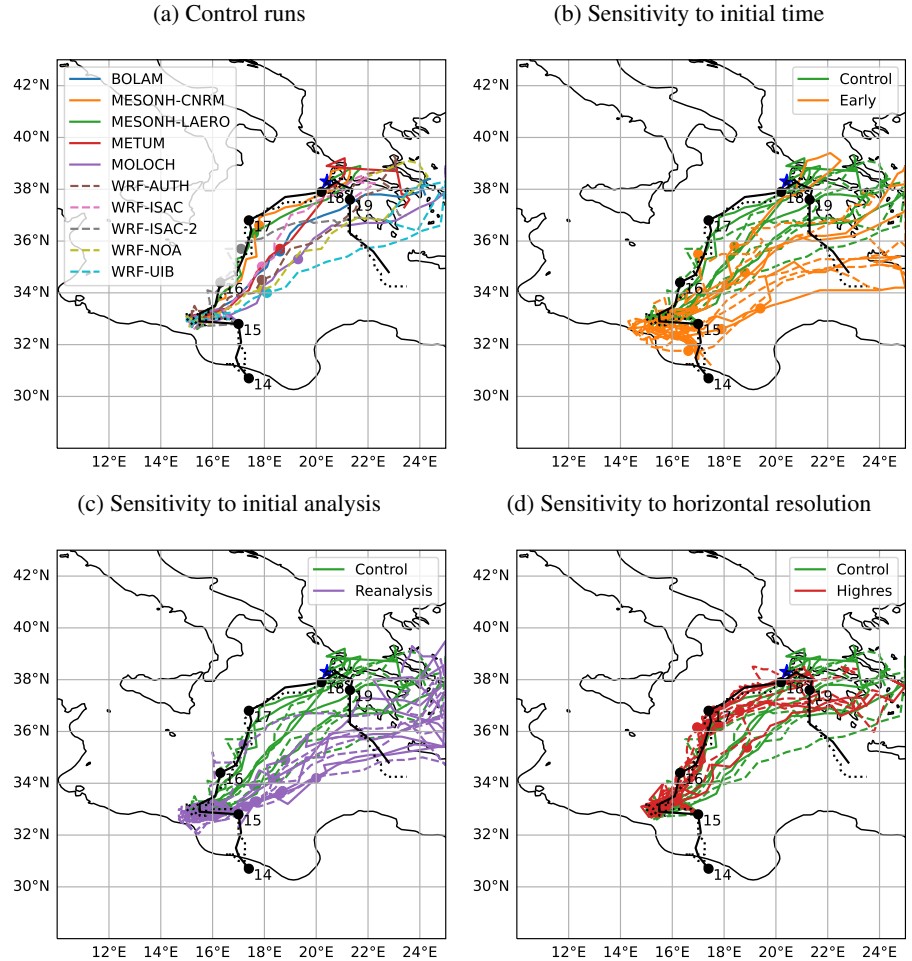

**Figure 4.** Track of Ianos (dashed curves for WRF and solid curves for other models) and location at 00 UTC 17 September (dots) in the control (a) and sensitivity runs to the initial time (b), initial data (c) and horizontal resolution (d). The control runs in (a) are plotted in green in (b–d). The IFS analyses, ERA5 reanalyses and the station of Palliki are overplotted as in Fig. 3.





**Figure 5.** As Fig. 4 but for timeseries of the intensity of Ianos.

(a) Control runs

(b) Sensitivity to initial time

(c) Sensitivity to initial analysis

(d) Sensitivity to horizontal resolution

**Figure 6.** As Fig. 4 but for timeseries of the 10-m wind speed averaged within 1 great circle degree from the simulated position of Ianos. Winds in the IFS analyses (black curves) and ERA5 reanalyses (dotted curves) are also shown for reference, as well as in the Advanced Scatterometer (ASCAT; blue diamonds) observations averaged around the position of Ianos in the IFS analysis.

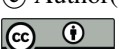

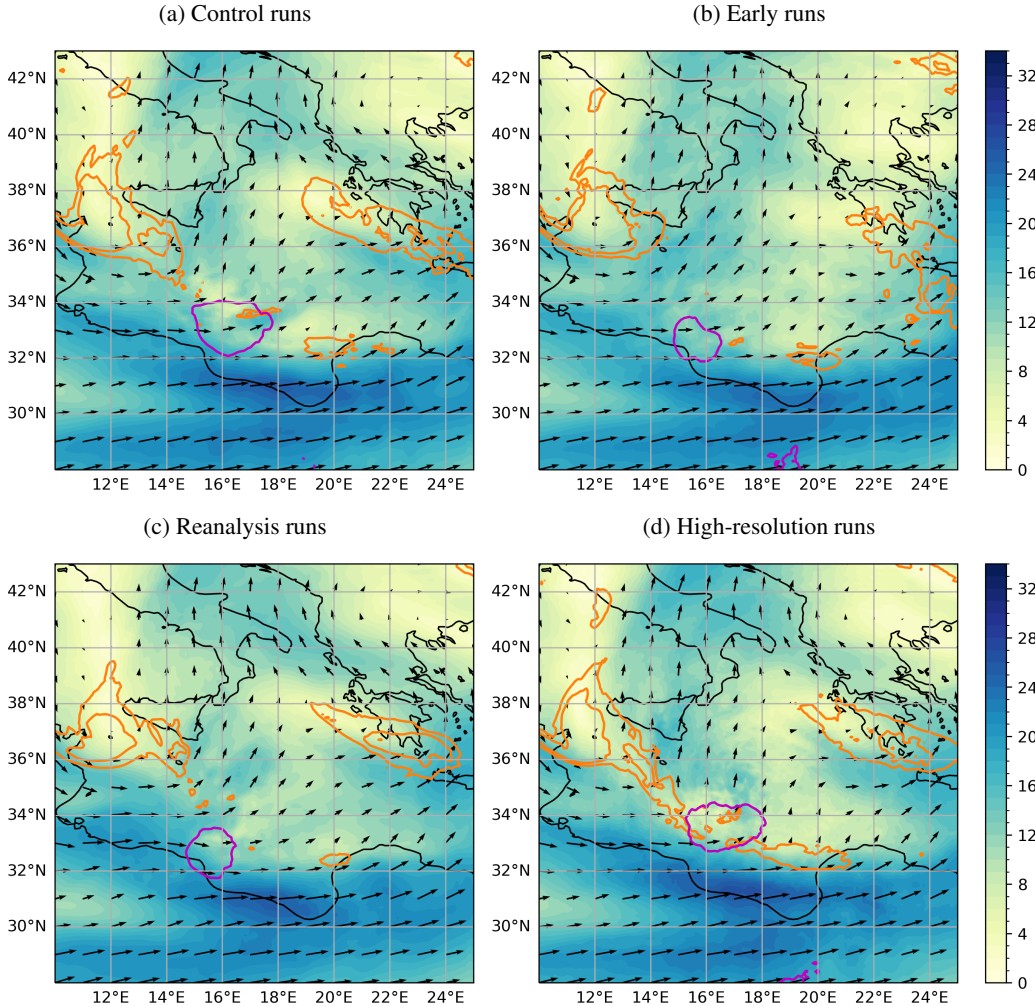

**Figure 7.** Representation of the phasing between Ianos and the large-scale dynamics in the model runs at 00 UTC 16 Sep 2020: as Fig. 2 but for composites of the control (a) and sensitivity runs to the initial time (b), initial analysis (c) and horizontal resolution (d). The 300 hPa potential vorticity is smoothed in (d) for clarity.



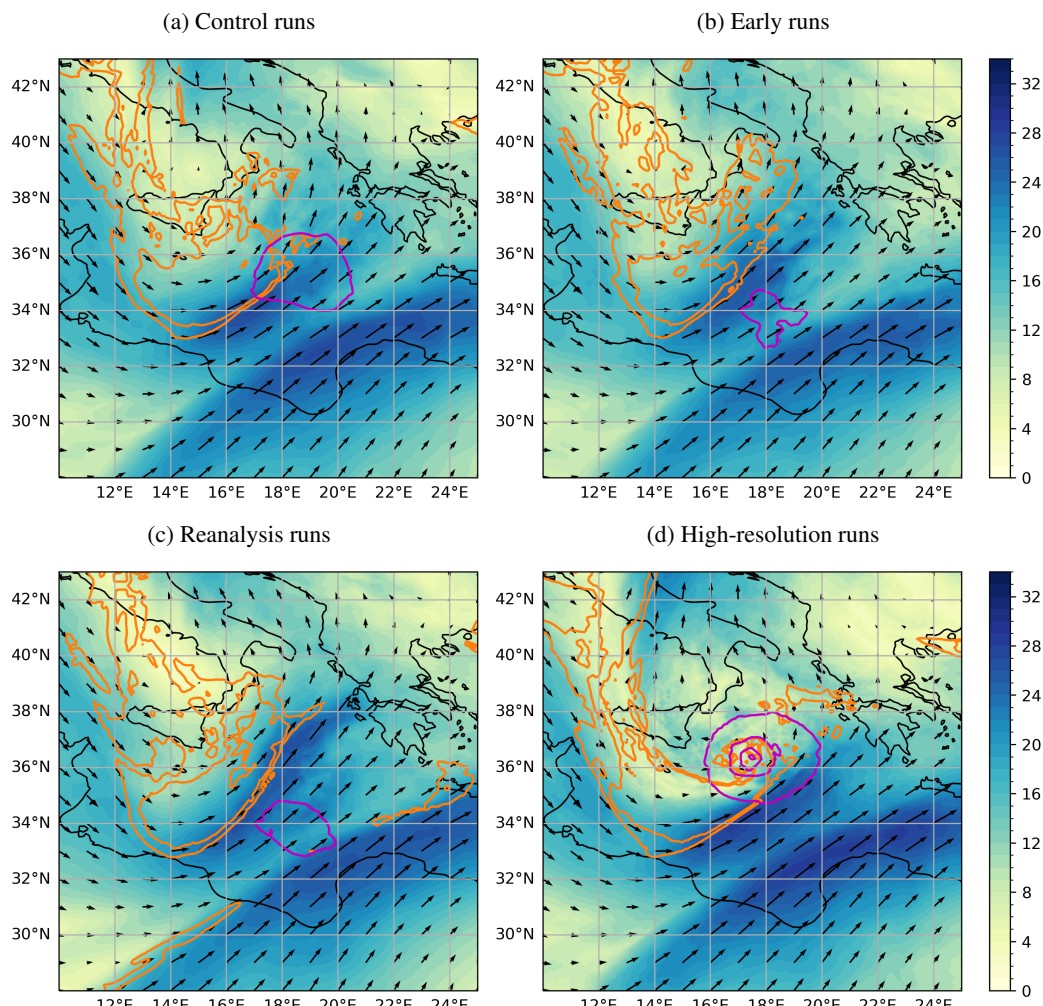

**Figure 8.** As Fig. 7 but at 00 UTC 17 Sep 2020.

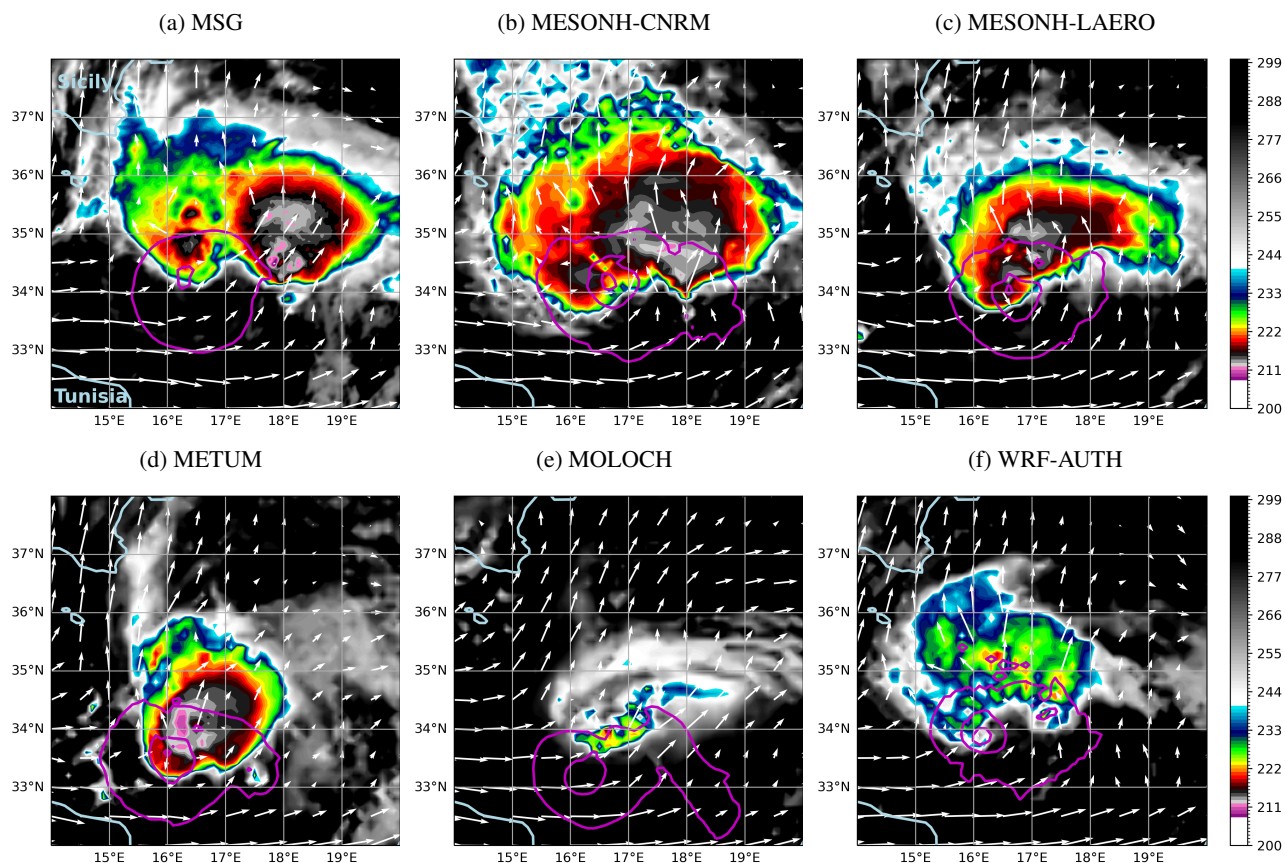

**Figure 9.** Infrared 10.8 μm brightness temperature (in K, low values highlighted) observed by Meteosat Second Generation (MSG, a) and simulated from high-resolution model runs (b–f) in an area of intense deep convection at 00 UTC 16 September. The 300 hPa wind (vectors) and MSLP (magenta contours every 5 hPa below 1010 hPa) are overlaid from the IFS analysis in (a) and from the high-resolution model runs in (b–f).





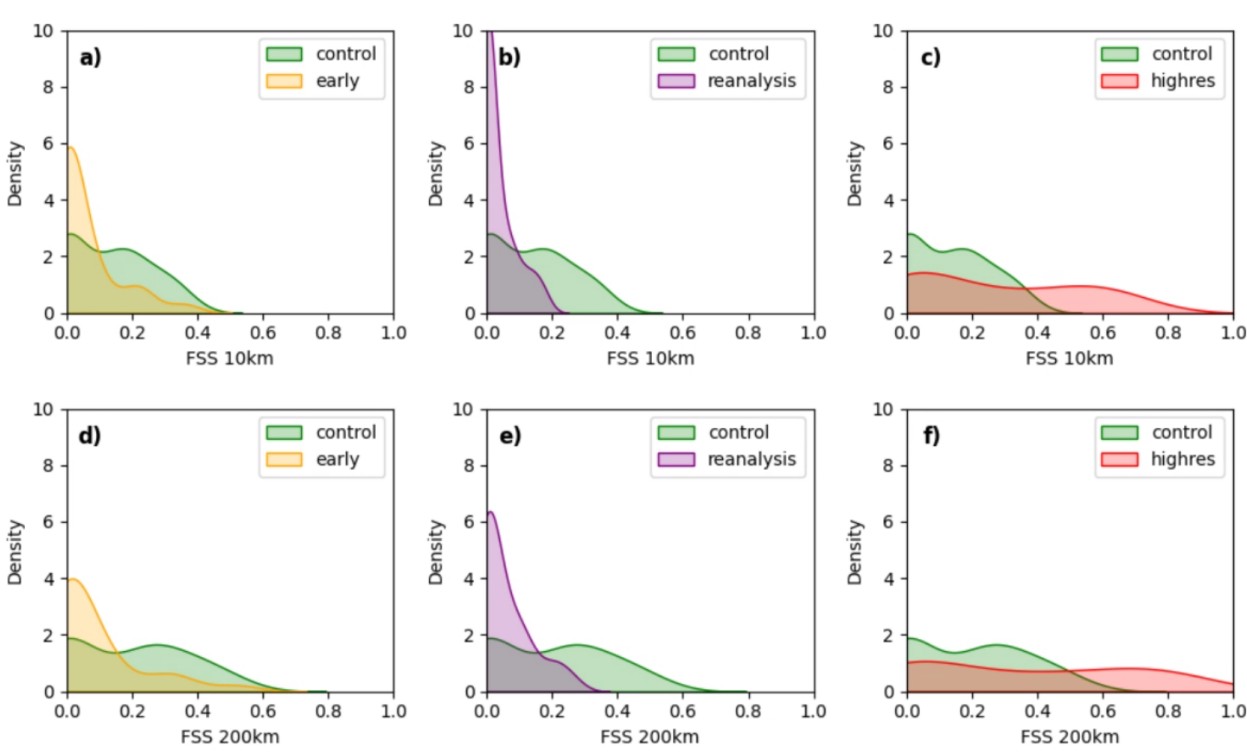

**Figure 10.** Probability distribution functions of the Fraction Skill Score (FSS) for the infrared 10.8 $\mu$m brightness temperature simulated in the control and sensitivity runs and assessed against MSG-SEVIRI observations from 18 UTC 15 September until 06 UTC 16 September for a threshold of 223 K and scales of 10 km (a–c) and 200 km (d–f) for the sensitivity to initial time (a,d), initial data (b,e) and horizontal resolution (c,f).



**Figure 11.** Timeseries of 3h accumulated precipitation along the track of Ianos in the control (a) and sensitivity runs to the initial time (b), initial analysis (c) and horizontal resolution (d). The precipitation is averaged within 1 great circle degree from the cyclone center in the model runs (coloured curves, dashed for WRF and solid for other models) and in GPM IMERG averaged around the position in the IFS analysis (black curves).



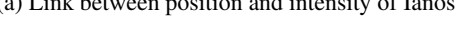

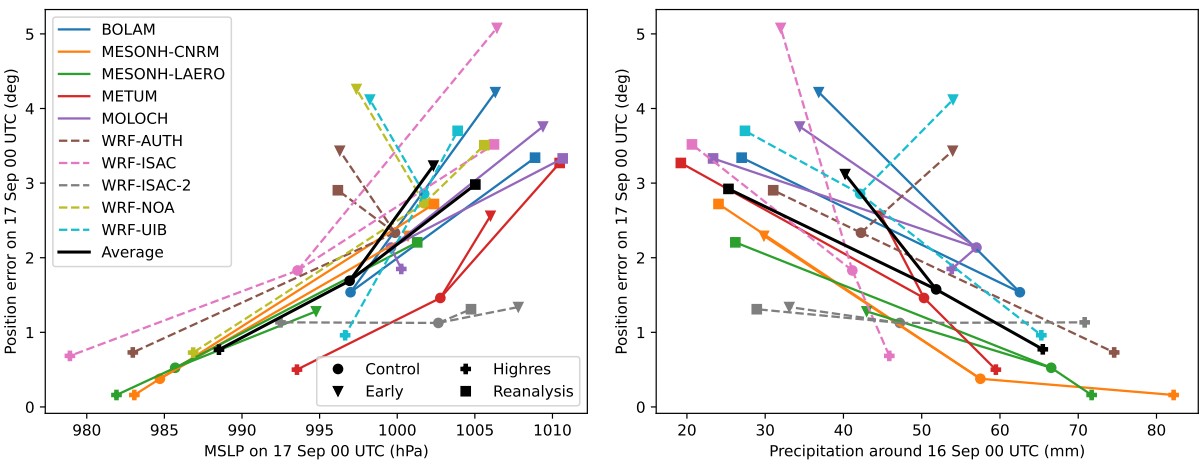

**Figure 12.** Summary of the sensitivity runs: error in the position of Ianos compared to the IFS analysis at 00 UTC 17 September (in great circle degree) as a function of the MSLP at 00 UTC 17 September (in hPa, a) and of the precipitation within 1 great circle degree along the track of Ianos from 18 UTC 15 September until 06 UTC 16 September (in mm, b). Symbols show the control runs (circles), early runs (triangles), reanalysis runs (squares) and high-resolution runs (pluses). Black symbols show averaged values over all 10 models and variants.





**Table 1.** Key characteristics of models and variants.

| Model | Grid spacing | Vertical levels | Deep convection scheme at 10 km |
|---|---|---|---|
| BOLAM | 10 km only | 60 levels | Kain (2004) |
| MESONH-CNRM | 10 and 2 km | 88 levels | Bechtold et al. (2001) |
| MESONH-LAERO | 10 and 2 km | 70 levels | Bechtold et al. (2001) |
| METUM | 10 and 2.2 km | 70 and 90 levels | Gregory and Rowntree (1990) |
| MOLOCH | 10 and 2 km | 60 levels | Kain (2004) also at 2 km |
| WRF-AUTH | 10 and 2 km | 50 levels | Kain (2004) |
| WRF-ISAC | 10 and 2 km | 57 levels | Kain (2004) |
| WRF-ISAC2 | 10 and 2 km | 50 levels | Janjic (1994) |
| WRF-NOA | 10 and 2 km | 50 levels | Kain (2004) |
| WRF-UIB | 10 and 2 km | 50 levels | Kain (2004) |

**Table 2.** Configuration of control and sensitivity model runs.

| Name | Initial time | Initial data | Grid spacing |
|---|---|---|---|
| Control | 15/00 | IFS | 10 km |
| Early | 14/12 | IFS | 10 km |
| Reanalysis | 15/00 | ERA5 | 10 km |
| High-resolution | 15/00 | IFS | 2 km |