# Peer review of "The crucial representation of deep convection for the cyclogenesis of medicane Ianos"

_EGUsphere, 2024_

## Author Comment (AC1)

**Response to Reviewer #1**

We thank the reviewer for their time and constructive comments. We have complied with most of the proposed changes. In the following, the comments made by the reviewer appear in black, while our replies are in blue.

The present manuscript provides a comprehensive overview of the intensification and convection formation of Medicane Ianos using different limited area models. While there are many articles related to Medicane Ianos, the reviewer recognizes that this manuscript represents a thorough investigation of the most important factors that could affect the formation and intensification of the medicane, including initial data, resolution, and initial time, with significant scientific collaboration.

The manuscript is very well written. Despite the complexity of the research, which involves numerous models and sensitivity experiments, the authors have managed to condense the information in the figures and text in a way that flows smoothly and is easy for readers to follow.

I believe the document should be published, with only a few minor comments that the authors should review or clarify. Otherwise, the document is excellent in its current state.

**MINOR COMMENTS:**

Line 190 (page 7): In the sentence "Additional simulations were initialized at 00 UTC on 14 September but are not shown, because they generally did not develop a cyclone," you use the word "generally," which does not imply "all." Does this mean that at least one model configuration developed a cyclone at 00 UTC on the 14th? If so, it would be good to be more specific, because if at least one model did develop a cyclone at that time, it would be worth describing here, as I don't know any previous scientific paper that showed that in the past.

In the large majority of models the initial surface low does not intensify and sticks to the Libyan coast. Only two models (WRF-AUTH and MESONH-LAERO) produce a cyclone but weaker and with larger bias in track compared to the later initializations. It was clarified that simulations initialised at 00 UTC on 14 September either produce even larger shift in track and underestimation in intensity or do not develop a cyclone (not shown). We prefer not discussing the individual model behaviour in more details, as we seek a robust response among models and variants.

Line 200 (page 7): Is there any database with estimated intensity based on different types of observations?

Unfortunately, there is no reliable satellite-based estimate of intensity for medicanes yet—such a the Dvorak technique for tropical cyclones. This has been clarified in the text.

Lines 323 - 325 (page 11): Here, you might consider discussing that explicit convection with deep convection parametrization switched OFF may result in overly strong cyclones and excessive precipitation in some cases. Additionally, you could mention that the ocean-atmosphere interaction in the model could help mitigate this negative effect.

We added a discussion on the apparent overestimation of wind and precipitation intensity at high resolution that might be mitigated by a more realistic representation of the complex air-sea interactions and feedbacks in this case, with reference to the recent paper by Karagiorgos et al. (2024) on coupled ocean-wave-atmosphere simulations of Ianos (though at lower resolution).

Figure 4: The legends in the figure captions should be described more thoroughly. While they are clear when reading the article, they are not immediately clear just from the figure caption.

We added a legend and clarified where to look at for the IFS analyses (black curves and dots at 00 UTC), ERA5 reanalyses (dotted curves) and the station of Palliki (blue star) to make the figure caption self-explanatory.

---

## Author Comment (AC2)

**Response to Reviewer #2**

We thank the reviewer for their time and constructive comments. We have complied with most of the proposed changes. In the following, the comments made by the reviewer appear in black, while our replies are in blue.

**Summary:**

This manuscript investigates the sensitivity of medicane Ianos to several model configurations. The analysis is based on several ensembles of simulations using 10 different models and different initial conditions, initialization times, and model grid spacing. Results show that relatively high resolution ( 2 km), accurate initial conditions, and a relatively short lead time are necessary to produce simulations with relatively small track and intensity error. Conversely, using relatively high resolution ( 10 km) with a cumulus parameterization or initializing the models too early results in relatively large track and intensity errors. The authors attribute this behavior to the phasing between a surface cyclone, its deep convection, and a nearby upper-tropospheric jet. This result leads to the conclusion that the representation of deep convection is important for the accurate representation of medicanes in numerical models, at least based on this case study.

**Evaluation:**

This study is the product of a major undertaking – at least 10 models were integrated with multiple configurations. This is not an easy task, and I commend the authors for doing a nice job at synthesizing the data from all those model simulations. I found the manuscript generally well written and well organized. I only found several areas where the writing could be improved (listed below). The manuscript's conclusions would have been even stronger with cyclone-relative composites that more clearly showed the deep convection and its phasing with the upper-tropospheric dynamics. Also, the manuscript could benefit from references to the ample body of work (outside the medicanes subdiscipline) that demonstrates the benefits of convection-permitting resolution for the representation of atmospheric phenomena.

We produced cyclone-relative composites as suggested and they confirm the sensitivity of the cyclone evolution to its phasing with the upper-tropospheric dynamics (see response to comment on L220–230 below). Note that Fig. 11 is also cyclone-relative as it displays precipitation within 1 great circle degree from the cyclone center to "remove the non-trivial sensitivity to the different cyclone positions" mentioned below. Also, selected references from the large body of literature have been added in the introduction and conclusions on the benefits of convection-permitting resolution for the representation of precipitation intensity and convective systems in the Mediterranean as well as cyclones in tropical regions.

**Minor/editorial comments:**

Lines 140–145: This may be a stylistic preference, but I wonder if you would consider narrating the evolution of Ianos in past tense since it was an observed event.

The narrative is now written in the past tense as suggested.

L150: I believe that "and surface low" should be "and a surface low"

Corrected

Figures: it would be helpful to your readers to have labels next to the color bars.

Added

L155–160: I'm not sure how to see the "baroclinic interaction" from Figure 2d. I know what you mean, but your readers may benefit from a more explicit description.

The sentence was rephrased to "The PV streamer wrapped around the surface cyclone on 17 September, indicating the end of baroclinic interaction".

L169: "are affected by large differences" is a confusing phrase. Would it be possible to rephrase for clarity?

Rephrased to "the IFS analysis and ERA5 reanalysis contrast with each other".

L171: "shallower MSLP" do you mean "stronger MSLP"? The word "shallower" is not appropriate here.

Rephrased to "While they both agree well with satellite observations on the cyclone track, the IFS analysis indicates MSLP values up to 10 hPa deeper than the ERA5 reanalysis during the period of maximum intensity."

Figures 4, 5, 6, & 11: The dashed lines in the panels with only two colors (panels b, c, & d) are unnecessary. These plots merge all models together and, as far as I understand it, they do not intend to highlight intermodal differences but sensitivities to initialization time, resolution, etc. Would you consider using only solid lines since it is difficult to differentiate between models anyway due to the single color for each sensitivity test?

Dashed lines have been changed to solid lines as suggested.

L180–185: Can the authors comment on the different WRF configurations and how they may have led to such wide spread?

We commented that "the choice of subgrid parameterization schemes can impact results as much as the choice of a mesoscale model (see Table 1 of the Supplementary material for key parameterizations of WRF model variants)".

L200–210: Your readers need more information about your ASCAT analysis. Did you average the ASCAT data within the same domain as in the models? Did you interpolate the ASCAT data to a similar grid as in your model for a clean comparison?

As the cyclone position differs between model runs, IFS analyses and ERA5 reanalyses, all wind data are averaged within 1 great circle degree from the cyclone position, including ASCAT observations. This provides a cyclone-relative perspective that makes results comparable and independent of the grid of the models or observations. We clarified in the caption of Fig. 6 that ASCAT observations are "averaged within 1 great circle degree from the position of Ianos in the IFS analysis". We now also refer to Fig. 2 of Ferrarin et al. (2023) for maps of observed wind fields on 16 and 17 September.

L206: I don't understand what "measured values" refers to. Please modify.

Clarified to "the values measured by ASCAT".

L207: "As for the MSLP" Did you mean "As in the MSLP"? I am confused. . .

Corrected to "As the MSLP".

L220–230: These statements would be more convincing if you could show cyclone-relative maps that would remove the non-trivial sensitivity to the different cyclone positions.

Figures R1-1 and R1-2 show the same variables as Figs. 7 and 8 of the paper but for cyclone-centered composites as suggested. By construction, the cyclone looks deeper than in the geographically-based composites. (As stated in the paper, "the much deeper cyclone in the high-resolution composite is due to both the deeper cyclone in individual simulations and the lower spread in cyclone position between the simulations.") Otherwise, Figs R1-1 and R1-2 essentially show the same sensitivity to the phasing of the cyclone with its environment as Figs. 7 and 8 of the paper, although with a more blurred double jet structure, thus confirm the statements. We prefer keeping the original figures, because they better depict the large-scale circulation and keep the geographical information.

L245: I don't understand this sentence. Please rewrite.

The sentence was removed, as the other simulations are not shown in Fig. 9 but discussed in the next paragraph based on Fig. 10.

Fraction skill score analysis and Fig. 10: The description of this analysis is insufficient to understand how you obtained your results. I am unsure how interpret Fig. 10 and even less confident on your readers' ability to replicate your analysis. Would it be possible to add more details about this analysis? Why do you obtain a PDF for each scale? Are you comparison on a grid point by grid point basis? On a model-by-model basis? More information would be very helpful here. Also, is it fair to start at a scale of 10 km when your coarsest model grid spacing is 10 km (i.e., you are considering a single grid point)? Why not a scale that is more representative of the effective model resolution (~6*dx)?

We clarified that "Results for two representative scales of 10 and 200 km are illustrated in Fig. 10 as probability distribution functions of the FSS over the models and variants, grid points and time steps". The caption of Fig. 10 has been updated accordingly. The scale of 10 km might not be fully fair for the 10 km runs but results are very similar for a scale of 50 km (see Fig. R1-3 for a range of scales). Thus, we prefer keeping the current scale of 10 km to avoid discussing the effective model resolution, which depends not only on the grid spacing but also on the model and numerical scheme.

L260-265: It looks like you used the 3-hourly IMERG product. Did you use the level 3 (research) product? I don't believe you mentioned this earlier. For a robust comparison, I would recommend interpolating the model output to the IMERG grid before averaging over a domain.

We use the level 3 product in version 07 (see reference Huffman et al. 2023 cited in the Methods). We clarified that "The original 30 min IMERG data is resampled to 3 h to facilitate comparison" and averaged "within 1 great circle degree from the position in the IFS analysis in IMERG (black curves)". The data is provided on the same regular 0.1°×0.1° grid as the output of model runs, thus no spatial interpolation is needed.

Figure 11: This figure needs a label for "IMERG" corresponding to the black line.

Added

L215: The attribution of wind overestimation to "an inaccurate representation of the air-sea interactions" is speculative; i.e., this manuscript does not show anywhere that the air-sea interactions are not properly represented in the model.

We agree it is a hypothesis but we do not believe it is speculative. As explained in the following sentence, "not all the models and variants properly take into account the increase and saturation in surface roughness due to developing and breaking waves, respectively, or the cooling of the sea surface, which may have complex and relevant feedbacks on the atmospheric boundary layer and surface winds". Indeed, our hypothesis seems to be confirmed by the recent paper by Karagiorgos et al. (2024) on coupled ocean-wave-atmosphere simulations of Ianos (though at lower resolution). We added a sentence in the conclusions on the representation of the complex air-sea interactions and feedbacks, with reference to this paper.

L275: This statement is purely speculative (i.e., you haven't shown sufficient proof): "Thus, intense convection during the second burst around 00 UTC on 16 September is crucial for the phasing between Ianos and the large-scale dynamics, which pilots the cyclone track and intensity via steering and baroclinic interaction."

We admit that the processes of upscale impact of convection are not fully elucidated in the paper but we do not see an alternative explanation to why convection-permitting resolution clearly improves the representation of the large-scale circulation, for which 10 km horizontal grid spacing is largely sufficient. We reworded to "appears" crucial for the phasing between Ianos and the large-scale dynamics.

L291: What is that "alternative scenario"? I admittedly don't know what this is referring to. Please be more explicit.

Clarified to "the alternative scenario with a cyclone that quickly intensifies close to the North African coast".

L290–295 & L340: It is true that MESONH performs relatively well. Coincidentally, this model has the highest number of vertical levels. I realize that testing the sensitivity to vertical grid spacing is beyond the scope of

this study, but this needs to be acknowledged. One could argue that vertical resolution is equally or even more important than horizontal resolution when resolving deep convection and its turbulent processes.

We added that "Apart from the horizontal resolution, the vertical resolution differs between the models and variants and may also influence the representation of deep convection." However, the influence is not straightforward as the increase in the number of vertical levels between MESONH-LAERO and MESO-CNRM does not clearly improve the models, whereas METUM has similar number of levels but very different results.

L330–335: I do not understand how the "sensitivity to both initial conditions and model physics suggests a balanced contribution of baroclinic and diabatic processes." Please be more specific.

Rephrased to "This interplay aligns with the balanced contribution of baroclinic and diabatic processes to the cyclone dynamics found in previous medicanes and other Mediterranean cyclones".

[Figure]

Figure R1-1: Representation of the phasing between Ianos and the large-scale dynamics in the model runs at 00 UTC 16 Sep 2020: as Fig. 7 of the paper but for cyclone-centered composites.

[Figure]

(a) Control runs  (b) Early runs

(c) Reanalysis runs  (d) High-resolution runs

**Figure R1-2**: Representation of the phasing between Ianos and the large-scale dynamics in the model runs at 00 UTC 17 Sep 2020: as Fig. 8 of the paper but for cyclone-centered composites.

[Figure]

**Figure R1-3**: Probability distribution functions of the Fraction Skill Score (FSS) as Fig. 10 of the paper but for a wider range of scales.